# Green Synthesis of Zinc Oxide Nanoparticles (ZnO-NPs) by *Pseudomonas aeruginosa* and Their Activity against Pathogenic Microbes and Common House Mosquito, *Culex pipiens*

**DOI:** 10.3390/ma14226983

**Published:** 2021-11-18

**Authors:** Abdullah M. Abdo, Amr Fouda, Ahmed M. Eid, Nayer M. Fahmy, Ahmed M. Elsayed, Ahmed Mohamed Aly Khalil, Othman M. Alzahrani, Atef F. Ahmed, Amal M. Soliman

**Affiliations:** 1Botany and Microbiology Department, Faculty of Science, Al-Azhar University, Nasr City, Cairo P.O. Box 11884, Egypt; abdullah-abdo@azhar.edu.eg (A.M.A.); aeid.microbiology@azhar.edu.eg (A.M.E.); khalilahmed_1980@hotmail.com (A.M.A.K.); 2Marine Microbiology Laboratory, National Institute of Oceanography and Fisheries, Cairo P.O. Box 101, Egypt; nm_fahmy@niof.sci.eg; 3Department of Anesthesiology, Intensive Care and Pain Management, Faculty of Medicine, Ain Shams University, Cairo P.O. Box 1181, Egypt; Ahmed-elsayed@med.asu.edu.eg; 4Department of Biology, College of Science, Taif University, P.O. Box 11099, Taif 21944, Saudi Arabia; o.alzahrani@tu.edu.sa (O.M.A.); atefali@tu.edu.sa (A.F.A.); 5Department of Medical Microbiology and Immunology, Faculty of Medicine, Ain-Shams University, Cairo P.O. Box 1181, Egypt; amal-soliman@med.asu.edu.eg

**Keywords:** green synthesis, ZnO-NPs, *Pseudomonas aeruginosa*, antimicrobial activity, larvicidal activity

## Abstract

The synthesis of nanoparticles by green approaches is gaining unique importance due to its low cost, biocompatibility, high productivity, and purity, and being environmentally friendly. Herein, biomass filtrate of *Pseudomonas aeruginosa* isolated from mangrove rhizosphere sediment was used for the biosynthesis of zinc oxide nanoparticles (ZnO-NPs). The bacterial isolate was identified based on morphological, physiological, and 16S rRNA. The bio-fabricated ZnO-NPs were characterized using color change, UV-visible spectroscopy, FT-IR, TEM, and XRD analyses. In the current study, spherical and crystalline nature ZnO-NPs were successfully formed at a maximum SPR (surface plasmon resonance) of 380 nm. The bioactivities of fabricated ZnO-NPs including antibacterial, anti-*candida*, and larvicidal efficacy were investigated. Data analysis showed that these bioactivities were concentration-dependent. The green-synthesized ZnO-NPs exhibited high efficacy against pathogenic Gram-positive bacteria (*Staphylococcus aureus* and *Bacillus subtilis*), Gram-negative bacteria (*Escherichia coli* and *Pseudomonas aeruginosa*), and unicellular fungi (*Candida albicans*) with inhibition zones of (12.33 ± 0.9 and 29.3 ± 0.3 mm), (19.3 ± 0.3 and 11.7 ± 0.3 mm), and (22.3 ± 0.3 mm), respectively, at 200 ppm. The MIC value was detected as 50 ppm for *E. coli*, *B. subtilis*, and *C. albicans*, and 200 ppm for *S. aureus* and *P. aeruginosa* with zones of inhibition ranging between 11.7 ± 0.3–14.6 ± 0.6 mm. Moreover, the biosynthesized ZnO-NPs showed high mortality for *Culex pipiens* with percentages of 100 ± 0.0% at 200 ppm after 24 h as compared with zinc acetate (44.3 ± 3.3%) at the same concentration and the same time.

## 1. Introduction

The widespread disease caused by pathogenic microbes and arthropod pests and vectors is considered the main challenge to discover new active compounds. The overuse of antibiotics and chemical pesticides led to the fast-growing resistant strains, as well as negative impacts on human health and the ecosystem. The linkage between nanotechnology and disease control science opens the way to limit the spread of these resistant strains [1,2]. Nanotechnology is a multidisciplinary field concerned with the manufacture, design, and application of nanoscale materials (1–100 nm). Due to the distinctive properties of nanomaterials, they have been applied in various fields such as materials science, chemistry, agriculture, medicine, physics, textile industry, and pharmaceuticals [3,4,5]. Contrary to a conventional concept, the nanoscale length affects the structure and total energy leading to changes in optical, thermal, mechanical, electronic, magnetic, chemical, and structural properties that characterize nanomaterial more than bulk materials [6,7]. 

Physical and chemical methods such as the sol-gel method, solvothermal synthesis, laser ablation, microwave-supported synthesis, and metal reduction by various reducing agents such as sodium citrate, hydrazine hydrate, and sodium borohydride are used for NPs’ synthesis [8,9]. However, these methods use harsh conditions as well toxic chemicals in the reduction and stabilization processes that limit their applicability. Therefore, biological methods or green methods are highly chosen because they are safe, economical, clean, and eco-friendly, and they are effective and easy sources for producing pure materials with high productivity. Moreover, the biological synthesis does not require dangerous and toxic compounds or high temperatures and pressures, in addition to avoiding the use of external reducing, capping, and stabilizing agents [10,11]. Various bacterial species, unicellular fungi, multicellular fungi, actinomycetes, and plant extracts as biological entities are utilized to fabricate various metal and metal oxide NPs [12,13,14,15]. Recently, agricultural wastes were used as a catalyst for the fabrication of different nanomaterials and utilized for purification and treatment of aqueous solutions containing various pollutants as a green approach [16]. Bacteria are the major prokaryotes’ group that exist in water, soil, plants, and animals, and are widely utilized for nanoparticles synthesis through extracellular or intracellular mechanisms. In the intracellular mechanism, the ions are mobilized into the bacterial cell and synthesis of NPs inside the bacterial cells, whereas the extracellular mechanism is used as biomass filtrate of the bacterial cell containing various secondary metabolites for fabrication of NPs [17]. 

Zinc oxide nanoparticles (ZnO-NPs) are characterized by unique chemical and optical properties, which distinguish them from the rest of the nanomaterials and make them of interest. It is a type of metal oxide with new catalytic and oxidizing properties that enable them to integrate into various biotechnological and biomedical applications [18]. ZnO-NPs have been used in a variety of cutting-edge applications including biology, cosmetics, pharmaceutical industry, animal food supplements, medical fields, environmental protection, coatings, rubber, communications, sensors, and electronics [3,19]. Due to their low toxicity and size-dependent properties, ZnO-NPs have been used in many fields such as microelectronics, textiles, cosmetics, and diagnostics. ZnO-NPs are recognized as safe and have antimicrobial properties that qualify them to treat many infectious diseases in humans and animals more broadly than other metals that are exploited for the synthesis of nanoparticles for clinical application [20]. In addition to their antifungal properties, the biogenic ZnO-NPs showed considerable larvicidal activity against various mosquitoes such as malaria vector, *Anopheles stephensi*, and *Aedes aegypti*, making them effective and safe candidates for insect control [21,22]. Bacteria are distinguished from the rest of the microbes used for the bio-fabrication of ZnO-NPs for their ease of processing and their manipulative genetic characteristics, which make them the best choice in this regard compared to eukaryotic microorganisms [23].

Mangrove forests are found at the intersection of marine and terrestrial habitats and are home to the biodiversity of microorganisms and plants. Microbes are one of the vital components of the mangrove ecosystem, have a key role in the formation and maintenance of this biosphere, and are a source of useful products in the field of biotechnology [24]. Therefore, the current study focused on the green synthesis of ZnO-NPs by using the bacterial strain *Pseudomonas aeruginosa* isolated from mangrove sediment. The bacterially fabricated ZnO-NPs were characterized by color change, UV-visible spectroscopy, Transmission Electron Microscopy (TEM), Fourier Transform Infrared (FT-IR) spectroscopy, and X-ray diffraction (XRD). Moreover, the inhibitory effect of biosynthesized ZnO-NPs was investigated against pathogenic Gram-positive bacteria, Gram-negative bacteria, and unicellular fungi as well as against the housefly *Culex pipiens*. 

## 2. Materials and Methods

### 2.1. Bacterial Isolation and Identification

The bacterial strain (coded as NMJ15) that was used for biosynthesis of zinc oxide nanoparticles (ZnO-NPs) was isolated from a mangrove sediment sample collected from Hurghada, Egypt (26°36′53″ N; 34°00′46″ E). The isolation was achieved by plating the sample onto nutrient agar media (containing g L^−1^: peptone, 5; beef extract, 3; NaCl, 5, agar, 15, distilled H_2_O, 1000 mL). 

The bacterial sample was primarily identified based on morphological, physiological, and biochemical tests according to a standard key. The identification was confirmed by amplification and sequencing of the 16S rRNA gene as follows: The genomic bacterial DNA was extracted according to Miller et al. [25] with a slight modification, while the PCR protocol was carried out according to Fouda et al. [26]. Briefly, a separate colony of bacterial species was picked up by using a sterilized toothpick, suspended in 50 µL of sterile H_2_O, and heated at 97 °C for 10 min in a water bath. The previously heated suspension was centrifuged (15,000 rpm for 10 min) and the supernatant, which contained the bacterial DNA, was collected. The intensity of DNA in the collected supernatant was measured by detecting its absorbance at 260 nm by using a UV spectrophotometer (Jenway 6305 UV spectrophotometer, 230 V/50 Hz, Staffordshire, UK ). The universal primers 27f (5′AGAGTTTGATCCTGGCTCAG-3′) and 1492r (5′GGTTACCTTGTTACGACTT-3′) were used to amplify the fragment of 16S rDNA. The PCR tube contained the following: 0.5 mM MgCl_2_, 1 × PCR buffer, 2.5 U Taq DNA polymerase (QIAGEN Inc., Germantown, MD, USA), 0.5 µM primers, 0.25 mM dNTP (Deoxynucleoside triphosphate), and approximately 5 ng of bacterial genomic DNA.

The PCR protocol was as follows: heating at 94 °C for 3 min, 30 cycles of 94 °C for a half min, 55 °C for a half min, and 72 °C for 1 min, followed by a final extension carried out at 72 °C for 10 min. The sequencing was achieved using an ABI 3730 × 1 DNA sequencer at GATC Company (Konstanz, Germany). The retrieved sequences were compared with those deposits on the Genbank database by using the NCBI BLAST software. The phylogenetic tree was constructed using bootstrap analysis. The sequences obtained in this study were deposited in Gene Bank under accession numbers MZ557381.

### 2.2. Bacterially Mediated ZnO-NPs Synthesis

The various metabolites secreted by bacterial strain NMJ15 were used as a reducing agent for zinc acetate dihydrate (Zn(CH_3_COO)_2_·2H_2_O) (Sigma Aldrich, Munich, Germany) as a precursor for ZnO-NPs. The bacterial strain was inoculated into nutrient broth media (pH 7) and incubated for 72 h at 35 ± 2 °C under shaking conditions (150 rpm). After an incubation period, the bacterial cells were harvested by centrifugation at 10,000 rpm for 5 min, followed by washing the collected cells thrice by distilled H_2_O to remove the attached medium components. Approximately 10 g of the collected bacterial cells were mixed with 100 mL distilled H_2_O and incubated at 35 ± 2 °C for 48 h. At the end of the incubation period, the previous mixture was subjected to centrifugation at 10,000 rpm for 5 min and we collected the supernatant or cell-free filtrate (CFF), which was used for biosynthesis of ZnO-NPs as follows: 100 mL of CFF (pH 8.5) was mixed with Zn(CH_3_COO)_2_·2H_2_O to get 2 mM as a final concentration and incubated at 35 ± 2 °C for 24 h under shaking condition. The formed creamy, white precipitate was harvested, washed twice with distilled H_2_O, and subjected to oven-drying (GESTER, 850W, Quanzhou, China) at 150 °C for 24 h. [27].

### 2.3. Characterization of Bacterially Synthesized ZnO-NPs

#### 2.3.1. UV-Vis Spectroscopy

The color change of CFF after being mixed with a precursor (Zn(CH_3_COO)_2_·2H_2_O) was monitored by UV-Vis spectroscopy at a wavelength of 200–600 nm. The color change was measured using a JENWAY 6305 Spectrophotometer (Jenway 6305 UV spectrophotometer, 230 V/50 Hz, Staffordshire, UK) to detect the maximum peak based on the surface plasmon excitation of formed ZnO-NPs. The bacterial CFF without zinc nitrate was used as a blank for the spectrophotometer. 

#### 2.3.2. Transmission Electron Microscopy (TEM)

The morphological characteristics of bacterially synthesized ZnO-NPs were investigated by TEM analysis (TEM-JEOL 1010, Tokyo, Japan). A few drops of the bacterially synthesized ZnO-NPs suspension were loaded onto the carbon–copper TEM grid. The excess ZnO-NPs’ solution on the TEM grid was removed by contacting the grid with blotting paper. The loaded TEM grid was left to dry at room temperature before being subjected to analysis [28].

#### 2.3.3. Fourier Transform Infrared (FT-IR) Spectroscopy

The role of bacterial metabolites in CFF in the reduction, capping, and stabilizing of ZnO-NPs was investigated using FT-IR analysis (Agilent system Cary 630 FT-IR model, Shimadzu, Tokyo, Japan). Approximately 0.2 g of biosynthesized ZnO-NPs was mixed well with potassium bromide (K Br) and subjected to high pressure to form a disk that was scanned at a wavenumber of 400 to 4000 cm^−1^.

#### 2.3.4. X-ray Diffraction (XRD) Patterns

The crystalline nature of bacterially synthesized ZnO-NPs was investigated using XRD analysis (X-ray diffractometer X’Pert Pro, Philips, Eindhoven, Netherlands). The operating conditions were θ range of 0°–80°, the X-ray radiation was Ni-filtered Cu Ka, and the current and voltage used were 30 mA and 40 kV, respectively. The crystal size of the biosynthesized ZnO-NPs was calculated based on XRD analysis using the Debye–Scherrer Equation as follows [29]:(1)$$\mathrm{Average}\;\text{ZnO-NPs}\;\mathrm{size}\;(D)=\frac{0.9\lambda }{\beta \cos\theta }$$
where 0.9 is the Scherrer’s constant, λ is the X-ray wavelength (1.540 Å), β is the half-width at half maximum (FWHM) peak, and θ is the Bragg’s angle. 

### 2.4. Biological Activity of Bacterially Synthesized ZnO-NPs

#### 2.4.1. Antimicrobial Activity

Antimicrobial activity of bacterially synthesized ZnO-NPs was investigated against pathogenic Gram-positive bacteria represented by *Staphylococcus aureus* ATCC6538 and *Bacillus subtilis* ATCC6633 and Gram-negative bacteria represented by *Pseudomonas aeruginosa* ATCC9022, *Escherichia coli* ATCC8739, and *Candida albicans* ATCC10231 as a unicellular fungus. To achieve this goal, the Mueller Hinton agar plates (already prepared, Oxoid) were inoculated with overnight bacterial and unicellular fungal strain (we adjusted the inoculum at O.D. = 1.0 for all tested organisms). One hundred microliters (100 µL) of NPs’ solution (200 ppm) were added to a well (0.8 mm) previously prepared in the inoculated plate. 

The activity of different ZnO-NPs’ concentrations (150, 100, 50, 25 ppm) was investigated to detect the MIC (minimum inhibitory concentration) value. The loaded plates were kept in the refrigerator for 2 hours before being incubated at 35 ± 2 °C for 24 h to confirm the successful spread of NPs’ solution in plates before bacterial growth. After the incubation period, the positive results were recorded as the diameter of the zone of inhibition (mm) formed around each well. The zinc acetate at the same concentrations was running with the experiment as a positive control, whereas Dimethyl sulfoxide (DMSO) (solvent system) was used as a negative control [29,30]. The experiment was carried out in triplicate.

#### 2.4.2. Larvicidal Bioassay

##### Insect Rearing and Experimental Conditions

The third instar larvae of *Culex pipiens* (housefly) was obtained from Medical Entomology Institute, Doki, Giza, Egypt. The experiment and mosquito rear were achieved in the Medical Entomology Lab., Department of Zoology and Entomology, Faculty of Science, Al-Azhar University, Cairo, Egypt. The experimental conditions were adjusted at 30–35 °C, relative humidity (RH) 65–80%, and light: dark condition of 12:12-h photoperiod. The larvae were grown on a plastic cup containing 500 mL tap water and fed on fish for 15 days. The weight of fish feeding per day was different according to larvae stage as follows: 0.1 g for first and second instar larvae, 0.3 g for third instar larvae, and 0.5 g for fourth instar larvae [31]. The experiment was achieved on the third instar larvae. 

##### Larvicidal Bioassay

The efficacy of bacterially synthesized ZnO-NPs as a nano-insecticide was evaluated against third instar larvae of housefly *C. pipiens*, as recommended by the World Health Organization [32]. Approximately 25 of the third instar larvae of *C. pipiens* were picked up from their colony using a pasture pipette and added to a plastic cup (500 mL) containing a 100-mL colloidal solution of bacterially synthesized ZnO-NPs (200 ppm). The same previous step was repeated with various ZnO-NPs and zinc acetate concentrations (150, 100, 50, and 25 ppm) (25 third instar larvae for each concentration). The cup containing tap water was run with the experiment as a control. Larvae were considered dead if there was no sign of any movement even after mild touch with a glass rod. The results were recorded after 24, 48, 72, and 96 h. The mortality percentages (%) were measured using the following Equation (2).
(2)$$\mathrm{Mortality}\;\mathrm{percentages}\;(\%)=\frac{\mathrm{Number}\;\mathrm{of}\;\mathrm{larvae}\;\mathrm{dead}}{\mathrm{Number}\;\mathrm{of}\;\mathrm{all}\;\mathrm{treated}\;\mathrm{larvae}}\;\times \;100$$

### 2.5. Statistical Analysis

Data in the current study were analyzed by Minitab^®^ 18.1program. A one-way analysis of variance (ANOVA) was performed on the data. Differences were considered significant when *p* < 0.05. The means of three replications and standard error (±SE) were also calculated for all the results obtained. 

## 3. Results and Discussion

### 3.1. Bacterial Isolation and Identification

The bacterial species isolated from mangrove sediment were characterized by their diverse medical and biotechnological applications. For example, culturable bacterial communities isolated from mangrove sediment were characterized by their efficacy in producing various enzymes such as amylase, chitinase, cellulase, phosphatase, urease, and protease [33]. Different bacterial species were isolated from mangrove sediments such as *Desulfovibrio*, *Azotobacter*, *Azospirillum*, *Klebsiella* sp., *Rhizobium*, *Bacillus*, *Xanthobacter*, *Enterobacter*, and *Pseudomonas* sp. [24]. In the current study, bacterial strain NMJ15 was isolated from mangrove sediment. The primary identification showed that the bacterial strain NMJ15 was Gram-negative, rod-shaped, motile, and positive for catalase, while its oxidase was negative. This strain can be grown in a wide temperature range (30–45 °C) and wide, broad pH (5–9). The bacterial identification was confirmed by amplification and sequencing of the 16S rRNA gene. Data represented in Figure 1 revealed that the bacterial strain was similar to *Pseudomonas aeruginosa* with a percentage of 99.68%.

### 3.2. Synthesis of ZnO-NPs Using the Bacterial Strain NMJ15

The fabrication of various metal and metal oxide NPs by the biological method has gained more interest over traditional chemical and physical methods. The biosynthesis or green synthesis has involved the utilization of various active compounds secreted by different biological beings (plants, actinomycetes, bacteria, yeast, and fungi) to reduce and stabilize new compounds at the nanoscale [9]. Green synthesis is characterized by biocompatibility and high productivity; being eco-friendly, inexpensive, and simple; not requiring harsh conditions; and avoiding the usage of toxic substances [34]. In the current study, the metabolites that existed in the cell-free filtrate of *P. aeruginosa* strain NMJ15 were used as a biocatalyst for forming ZnO-NPs. At first, a creamy-white precipitate as ZnO·H_2_O (hydrate zinc oxide) was formed due to the mixing of bacterial CFF with a precursor under stirring conditions. After that, the as-formed white precipitate was harvested by centrifugation and subjected to calcination at a high temperature (150 °C) for 24 h (Equations (3) and (4)).
(3)$$\mathrm{Zn}{\left({\mathrm{CH}}_{3}\mathrm{COO}\right)}_{2}\cdot 2H_{2}O+\mathrm{CFF}\overset{\mathrm{Stirring}}{\to }\mathrm{ZnO}\cdot H_{2}O(\mathrm{white}\;\mathrm{preciptate})$$
(4)$$\mathrm{ZnO}\cdot H_{2}O\overset{150\;{}^{\circ}C}{\underset{24\;h}{\to }}\text{ZnO-NPs}$$

The physicochemical characteristics of the synthesized NPs such as crystalline nature, size, and shape were controlled according to the microbes that were utilized in the fabrication process [35]. In our previous study, a creamy white precipitate of hydrated zinc oxide (ZnO·H_2_O) was formed after mixing the zinc acetate with a biomass filtrate of *Fusarium keratoplasticum* A1-3 and *Aspergillus niger* strain G3-1, which was subjected to calcination at 120 °C for 12 h to form ZnO at the nanoscale [19]. The ZnO-NPs were synthesized by various biological entities to integrate into biomedical applications. For instance, ZnO-NPs fabricated by water extract of *Sargassum wightii* exhibited antimicrobial activity and antibiofilm and insecticidal properties [36]. Additionally, ZnO-NPs synthesized through harnessing metabolites of bacterial species, *Serratia nematodiphila*, showed high antimicrobial and photocatalytic activities [29]. To date, the current study is the first report for the biosynthesis of ZnO-NPs through harnessing the metabolites secreted by *P. aeruginosa* isolated from mangrove sediment. 

### 3.3. Characterization of Bacterially Synthesized ZnO-NPs

#### 3.3.1. UV-Vis Spectroscopy

The successful bacterial synthesis of ZnO-NPs was confirmed by the CFF color change from colorless to white precipitate. This change was checked by UV-Vis spectroscopy at a wavelength of 200–800 nm to detect the maximum surface plasmon resonance (SPR). The highest SPR was observed at 380 nm, which indicated the successful transformation of metal precursor (Zn(CH_3_COO)_2_·2H_2_O) to the final product (ZnO-NPs) (Figure 2). Compatible with the obtained data, Jain and co-authors reported that the maximum absorbance observed in the UV chart of ZnO-NPs synthesized by *Serratia nematodiphila* ZTB15 was at 379 nm as an indicator for a successful process [29]. Additionally, the maximum absorption band for ZnO-NPs fabricated by *Sphingobacterium thalpophilum* was at 379 nm [37]. The SPR was formed due to electrons’ oscillation from the valence band to conduction band after exposure to light at a particular wavelength [38,39]. Pachaiappan et al. [39] reported that the variation in maximum SPR for ZnO-NPs can be attributed to the type of precursor and calcination temperature. Who reported that the highest absorption peaks for ZnO-NPs were 334 nm, 338 nm, and 361 nm for a precursor of zinc acetate, zinc sulfate, and zinc nitrate, respectively. 

#### 3.3.2. Transmission Electron Microscopy (TEM)

The size and shape of biosynthesized ZnO-NPs were investigated by using TEM analysis. Based on Figure 3A, it can be concluded that the particles in a synthesized sample were well arranged, homogenous, and with spherical shapes and sizes ranging between 6 nm to 21 nm. Moreover, the size distribution of synthesized NPs based on the TEM image was shown in Figure 3B with an average diameter of 14.95 ± 3.5 nm (Figure 3B). Based on published studies, there was a strong correlation between the size of NPs and their activity; the activity increased as NPs’ sizes decreased [3,9,40]. Moreover, Farzana et al. [41] showed that the smaller ZnO-NPs had a greater inhibitory effect on the multi-drug-resistant pathogenic bacteria. Therefore, we predicted the high activity of ZnO-NPs synthesized in the current study due to their smaller size. Similarly, the TEM image of ZnO-NPs synthesized by alkalophilic bacterial strain *Alkalibacillus* sp. W7 exhibited the successful formation of spherical shape with a size range of 1 to 30 nm and an average particle size of 17 ± 1 nm [12]. 

#### 3.3.3. Fourier Transform Infrared (FT-IR) Spectroscopy

The role of functional groups that exist in the bacterial CFF in the reducing, capping, and stabilizing of NPs was investigated by FT-IR. The bacterial CFF showed nine intense peaks at 3244, 1646, 1501, 1446, 1401, 1225, 1089, 886, and 769 cm^−1^ (Figure 4A). The strong broad peak at 3244 cm^−1^ corresponded to stretching O–H, whereas the peak at 1646 cm^−1^ was related to stretching the C=O vibration protein-peptide bond [42]. The peaks at 1501 cm^−1^, 1446 cm^−1^, and 1401 cm^−1^ corresponded to stretching C–N vibration mode of aliphatic and aromatic amines that referred to the presence of proteins in CFF [43]. Moreover, the peaks at 886 cm^−1^ and 769 cm^−1^ referred to stretching amide IV for proteins [44]. These peaks were shifting during the biosynthesis of NPs. The FT-IR for ZnO-NPs (Figure 4B) showed 520, 630, 895, 1032, 1420, 1620, 2503, 2590, 3110, 3330, 3620, and 3700 cm^−1^. The peaks at wavenumber between 3000 to 3700 cm^−1^ referred to stretching O–H of carboxylic acid, and the vibration mode of the –OH group overlapped with stretching NH of the amines [45,46]. The weak peak at 2590 cm^−1^ referred to the stretching SH thiol group. The medium peak observed at 1620 cm^−1^ was related to bending primary amine (NH) overlapped with carboxylate salts or amide [47]. Moreover, a medium peak at 1430 cm^−1^ signified stretching C=O of carboxylic salts and was referred to as the adsorption of CO_2_ and carbonates (CO_3_^2−^) at the NPs’ surface [45]. The strong peaks at 1032 cm^−1^ and 895 cm^−1^ may be corresponding to stretching S=O of sulfoxide and bending C=C of alkene [48,49,50]. The successful formation of ZnO-NPs was confirmed at peaks of 400 to 700 cm^–1,^ as reported previously [21,51,52]. The FT-IR analysis of ZnO-NPs confirmed the involvement of different groups such as C=O, O–H, NH, and SH thiol groups present in bacterial CFF in the reducing, capping, and stabilizing processes. 

#### 3.3.4. X-ray Diffraction (XRD) Pattern

The crystalline nature phase of the synthesized ZnO-NPs was investigated by XRD. The XRD pattern (Figure 5) showed eight intense diffraction peaks (100), (002), (101), (102), (110), (103), (112), and (201) at 2θ values of 31.5°, 34.2°, 36.3°, 47.3°, 56.4°, 62.7°, 67.3°, and 96.1°, respectively. The observed peaks in the XRD pattern were matched with data reported in the Joint Committee on Powder Diffraction Standards (JCPDS, card No. 89-7102) for the crystalline nature of ZnO-NPs. Our data were compatible with the XRD pattern for the crystalline nature of biologically synthesized ZnO-NPs [12,53]. The average ZnO-NPs’ size corresponding to the maximum diffraction peak (101) was calculated using the Debye–Scherrer Equation and it was 21 nm (compatible with TEM analysis). The reason for the small observed peaks at various 2 theta values can be related to the crystallization of bacterial metabolites such as proteins and organic substances that coated the ZnO-NPs’ surface, as reported previously [54].

### 3.4. Biological Activity of Bacterially Synthesized ZnO-NPs

#### 3.4.1. Antimicrobial Activity

Zinc oxide nanoparticles have attracted more interest due to it their low toxicity, biocompatibility, and low-cost nanomaterial. Therefore, they can be integrated into various medical applications, for instance, antifungal, antibacterial, anticancer, antidiabetic, antioxidant, anti-insect, anti-inflammatory, drug delivery, and bioimaging applications [3,55]. Herein, the inhibitory effect of the bacterially mediated green synthesis of ZnO-NPs against pathogenic microbes was assessed by the agar well diffusion method compared with bulk material (Zn(CH_3_COO)_2_·2H_2_O) as a positive control and DMSO (solvent system) as a negative control. To attain this goal, pathogenic microbial strains represented by *Bacillus subtilis*, *Staphylococcus aureus*, *Escherichia coli*, *Pseudomonas aeruginosa*, and *Candida albicans* were used. The bulk material showed moderate activity against pathogens of *B. subtilis, P. aeruginosa*, *E. coli,* and *C. albicans* with zones of inhibition of 15.3 ± 0.3, 11.7 ± 0.2, 13.3 ± 0.3, and 15.6 ± 0.3 mm and 12.7 ± 0.3, 0.0 ± 0.0, 10.7 ± 0.6, and 13.7 ± 0.3 mm for 200 ppm and 150 ppm, respectively. Interestingly, the DMSO as negative control did not show any inhibitory action against the tested pathogenic microbes. Interestingly, the activity of ZnO was increased at the nanoscale, which was dose-dependent. Our results are well-suited with published studies, which reported that the activity of Ag-NPs, CuO-NPs, and MgO-NPs was higher than those reported by bulk material (AgNO_3_, CuSO_4_·5H_2_O, and Mg(NO_3_)_2_⋅6H_2_O) [54,56,57]. At a concentration of 200 ppm, ZnO-NPs showed the highest antimicrobial activity with ZOIs of 12.3 ± 0.9, 29.3 ± 0.3, 19.2 ± 0.3, 11.7 ± 0.2, and 22.3 ± 0.3 mm for *S. aureus*, *B. subtilis*, *E. coli*, *P. aeruginosa*, and *C. albicans*, respectively (Figure 6). The ZOIs were decreased by reducing the ZnO-NPs’ concentration. For example, the ZOIs formed due to treatment with 150 ppm of bacterial ZnO-NPs were 21.7 ± 0.2, 17.7 ± 0.3, and 18.3 ± 0.3 mm against *B. subtilis, E. coli,* and *C. albicans,* respectively; these ZOIs decreased to 14.6 ± 0.5, 11.5 ± 0.6, and 13.7 ± 0.4 mm at 50 ppm for the same pathogenic microbes (Figure 6). Our results were in good agreement with published studies that clarified that the green-synthesized ZnO-NPs have broad-spectrum activity against various pathogenic microbes including *S. aureus*, *E. coli*, *P. aeruginosa*, *B. subtilis*, *Mycobacterium tuberculosis*, *Klebsiella aerogenes*, *Proteus mirabilis*, *Xanthomonas oryzae*, *Alternaria* sp. *Aspergillus* sp., *Penicillium expansum*, and *C. albicans* [29,42,58]. The inhibitory effect of bacterially synthesized ZnO-NPs in the current study can be attributed to the small ZnO-NPs’ sizes (6–21 nm). Sharma and co-author showed that the maximum inhibitory effect (94.05%) was achieved due to treatment with a small ZnO-NPs’ size (12–30 nm) compared with a large size at the same concentration [18]. 

ZnO-NPs have an excellent antimicrobial action due to their unique physical and chemical characteristics at the nanoscale. This superior activity can be related to three different mechanisms: (1) release of toxic ions (Zn^2+^) inside the cell; (2) electrostatic attraction between ZnO-NPs and the bacterial cell wall, and (3) production of reactive oxygen species (ROS) [59]. The release of toxic ions represented by Zn^2+^ as a result of the dissolution of ZnO-NPs inside the cell can inhibit different bacterial cell functions such as enzyme activity, cell metabolism, proton motive force, active transport, and, ultimately, bacterial cell death [60]. The liberation of this toxic ion is dependent on the size and shape of NPs. Chang and co-authors reported that the spherical and small size of ZnO-NPs and CuO-NPs release higher concentrations of toxic ions (Zn^2+^) than those released by rod shape [61]. This phenomenon can be attributed to the smaller surface area of the spherical shape and, hence, increased equilibrium solubility. The production of ROS is followed by high oxidative stress and, hence, cell death or cell damage. The toxic effects due to ROS formation are attributed to the generation of various reactive species such as O^2–^ (superoxide anion), OH^–^ (hydroxyl ion), and H_2_O_2_ (hydrogen peroxide). The previous reactive species were generated due to the reaction of NPs with the OH group and absorbed H_2_O to form OH^–^ and H^+^ that consequently formed O^2–^. After that, the HO^2^ (generated from reacting of O^2–^ with H^+^) was formed, which subsequently reacted with electron and H^+^ to form H_2_O_2_ [62]. The final product (H_2_O_2_) penetrated the microbial cell wall and cell membrane to interact with lipids, proteins, and nucleic acid, leading to cell death. Interestingly, the O^2–^ and OH^–^ remained on the surface of the bacterial cell and could not penetrate the cell membrane due to their negative charge [10]. The electrostatic attraction between positively charged NPs and negatively charged bacterial cells was considered another mechanism for the inhibitory effect of ZnO-NPs. Due to this interaction, ZnO-NPs penetrated the microbial cell, distorting the plasma membrane structure and bacterial cell wall integrity that led to the discharge of cellular components [63,64]. 

The lowest concentration of active compounds that inhibit microbial growth is defined as MIC and it should be detected for each compound that can be integrated into biomedical applications [65]. Therefore, analysis of variance revealed that the MIC value for bacterially synthesized ZnO-NPs was 200 ppm for *S. aureus* and *P. aeruginosa*, while it was 50 ppm for *B. subtilis*, *E. coli*, and *C. albicans* (Figure 6). In the current study, the synthesized ZnO-NPs were more active against *B. subtilis*, *E. coli*, and *C. albicans*, more than *S. aureus* and *P. aeruginosa* at a low concentration. Our results are in complete agreement with published studies [27,52].

#### 3.4.2. Larvicidal Activity

The efficacy of bacterially synthesized ZnO-NPs to act as an insecticide was investigated against housefly *Culex pipiens* as compared with bulk material (Zn(CH_3_COO)_2_·2H_2_O) at intervals times (24, 48, 72, and 96 h). At low concentrations (25 and 50 ppm), the bulk material did not exhibit any activity with time, whereas ZnO-NPs exhibited mortality percentages for third instar larvae of 30 ± 0.0% and 40 ± 5.8% after 24 h and the mortality increased with time to reach 46.7 ± 3.3% and 56 ± 0.0% after 96 h. The mortality percentages caused by zinc acetate and ZnO-NPs were increased with concentration. Our results are in agreement with the data of Roopan et al. [51], who showed that the mortality percentages caused by ZnO-NPs against *Aedes aegypti* were 21.4 ± 2.3%, 35.2 ± 3.6%, 59.6 ± 5.2%, and 88.6 ± 1.2% for concentrations of 15, 30, 60, and 120 ppm, respectively. In the current study, the maximum mortality rate caused by zinc acetate was achieved for a concentration of 200 ppm; it was 44.3 ± 3.3% after 24 h and increased to 63.3 ± 3.3% after 96 h. In contrast, ZnO-NPs at a concentration of 150 ppm caused mortality of 85 ± 5.8% after 24 h and increased to 96 ± 3.3% after 96 h. Interestingly, 100% mortality was attained after 24 h of treatment with 200 ppm of bacterially synthesized ZnO-NPs (Table 1). The obtained results are compatible with those reporting that the larvicidal efficacy of zinc acetate is little on mosquito larvae as compared with ZnO-NPs [36]. Moreover, Velsankar and co-authors highlighted that the ZnO-NPs fabricated by seed aqueous extract of Cucurbita had the efficacy to inhibit 50% of fourth instar larvae of *Culex tritaeniorhynchus* at a concentration of 44.68 ppm and inhibited 90% at 119.9 ppm [22]. To date, the current study is considered the first report to assess the efficacy of bacterially synthesized ZnO-NPs as an insecticidal agent against housefly *Culex pipiens.*


The toxicity of NPs on mosquito larvae can be attributed to their efficacy to penetrate through the insect’s cuticle and, hence, destroy the respiratory openings on the skin followed by suffocation and ultimately to death [66]. Moreover, due to NPs’ penetration, ZnO-NPs can induce various morphological changes on the hemocytes upon exposure, which lead to deformations and decrease their viability [21]. Additionally, the toxicity of ZnO-NPs to insects can be related to their efficacy in inducing ROS and the liberation of toxic ions inside the insect’s cells [2]. Another larvicidal mechanism of ZnO-NPs can be related to their surface defects for a different texture [21]. These surface defects, especially at the corners and edges, can be abrasive and, hence, damage the larvae cell membrane. For instance, the needle and star shape of ZnO-NPs have more larvicidal efficacy through cutting the larval body, damaging the insect’s midgut, and destroying the respiratory system [1]. 

## 4. Conclusions

In the current study, ZnO-NPs were successfully green synthesized using bacterial strain *Pseudomonas aeruginosa* NMJ15. The cell-free filtrate (CFF) was used as a biocatalyst for capping and stabilizing ZnO-NPs. The bio-fabrication was confirmed by a color change of CFF from colorless to white precipitate. The color change was monitored by UV-Vis spectroscopy and showed maximum surface plasmon resonance at 380 nm, which was the characteristic peak of ZnO-NPs. The physicochemical characterization of as-formed ZnO-NPs was analyzed by TEM, FT-IR, and XRD analysis. Data showed the successful formation of the spherical and crystalline ZnO-NPs. The FT-IR chart exhibited the role of various functional groups, for instance, C=O, O–H, NH, and SH thiol groups that exist in CFF in the reduction of Zn ion to form ZnO-NPs. The biological activity of bacterially synthesized ZnO-NPs, including antimicrobial activity and larvicidal activity, was investigated. The data analysis showed that the efficacy of ZnO-NPs to inhibit the growth of pathogenic microbes with zones of inhibition of 12.3 ± 0.9, 29.3 ± 0.3, 19.2 ± 0.3, 11.7 ± 0.2, and 22.3 ± 0.3 mm for *S. aureus*, *B. subtilis*, *E. coli*, *P. aeruginosa*, and *C. albicans,* respectively, at 200 ppm. The activity of biosynthesized ZnO-NPs as bactericidal was dose-dependent. Moreover, the highest mortality (100%) of *Culex pipiens* was attained after 24 h due to 200 ppm ZnO-NPs. The investigated biological activities exhibited the high efficacy of bacterially synthesized ZnO-NPs as compared with bulk material (Zn(CH_3_COO)_2_·2H_2_O). The obtained results provide a promising tool for controlling the growth of pathogenic microbes as well as housefly mosquitos with the eco-friendly and cost-effective active substance.

## Figures and Tables

**Figure 1 materials-14-06983-f001:**
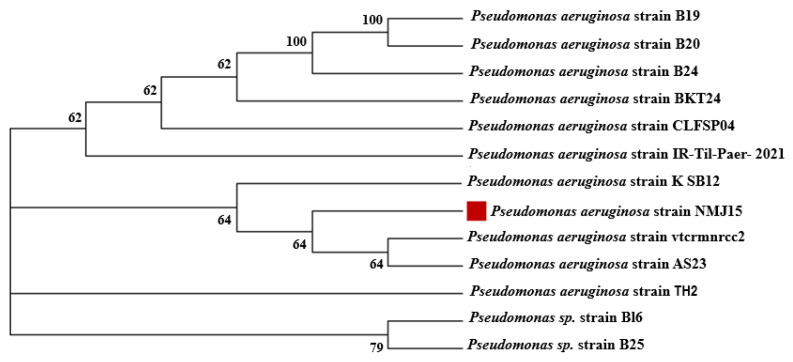
Phylogenetic tree of the 16S rRNA sequences of bacterial strain NMJ15 isolated from mangrove sediment. The analysis was achieved by MEGA 6, which used the neighbor-joining approach with a bootstrap value (1000 replicates).

**Figure 2 materials-14-06983-f002:**
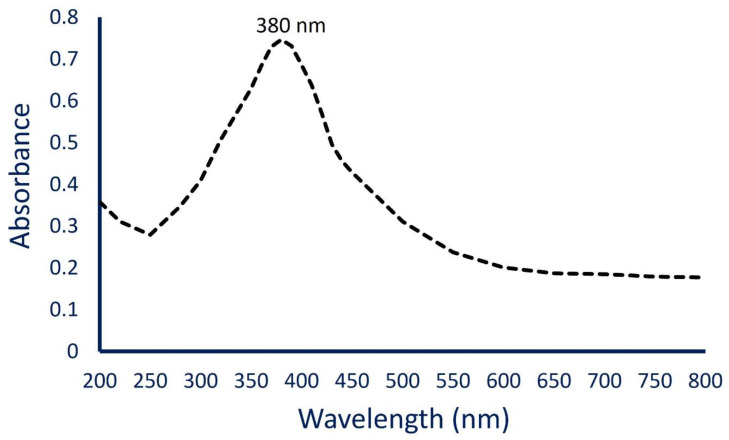
The UV-Vis spectroscopy for bacterially synthesized ZnO-NPs showed a maximum SPR at 380 nm.

**Figure 3 materials-14-06983-f003:**
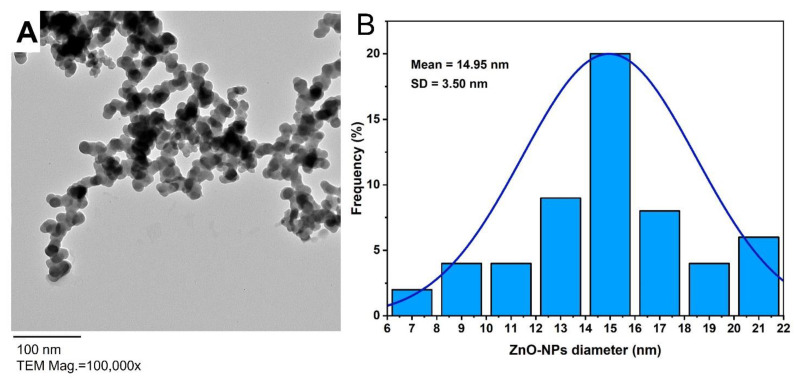
(**A**) is the TEM image for ZnO-NPs synthesized by bacterial strain *P. aeruginosa* NMJ15, (**B**) is the histogram that showed ZnO-NPs’ size distribution based on the TEM image.

**Figure 4 materials-14-06983-f004:**
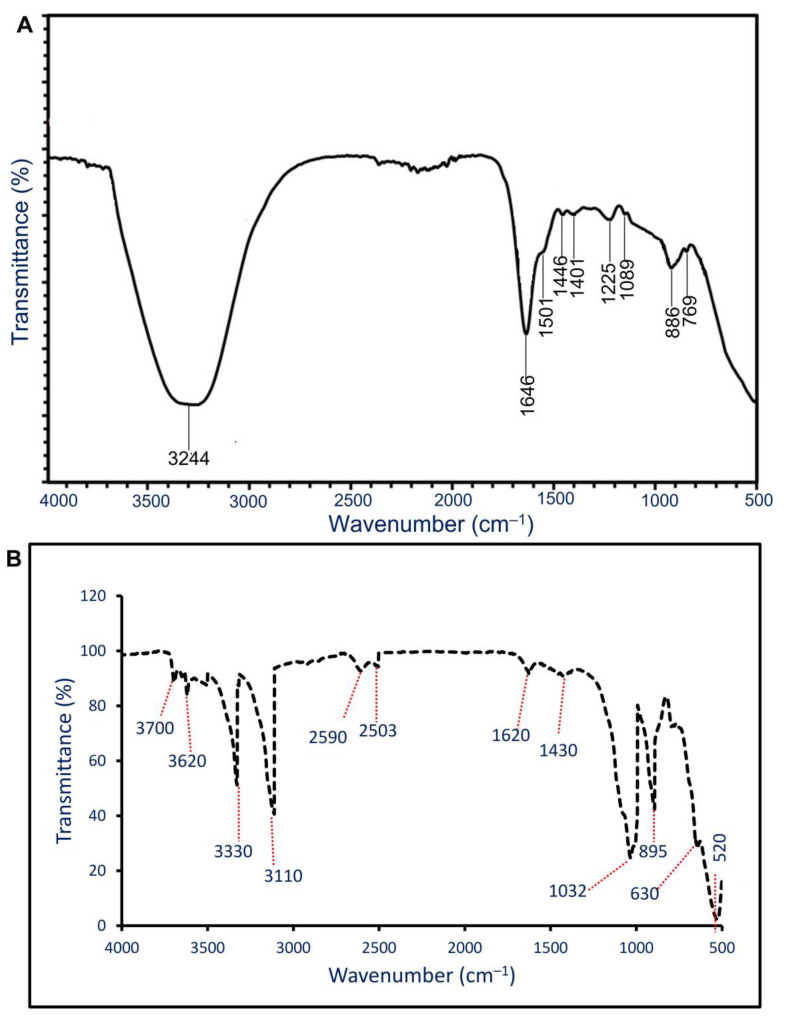
FT-IR of bacterial cell-free filtrate (**A**) and biosynthesized ZnO-NP (**B**).

**Figure 5 materials-14-06983-f005:**
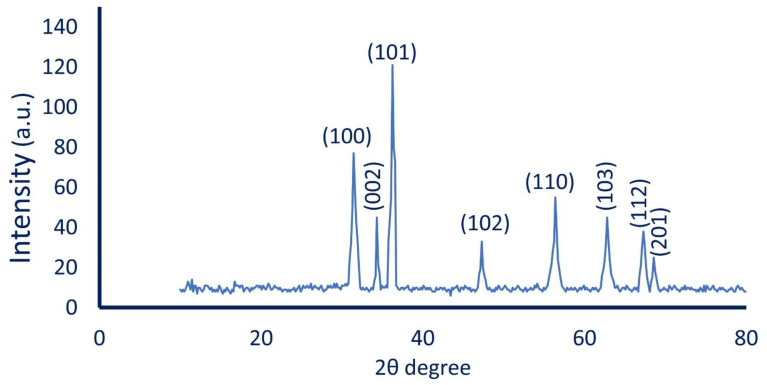
X-ray Diffraction (XRD) Patterns for bacterially synthesized ZnO-NPs showed the highest diffraction peaks at different 2θ values.

**Figure 6 materials-14-06983-f006:**
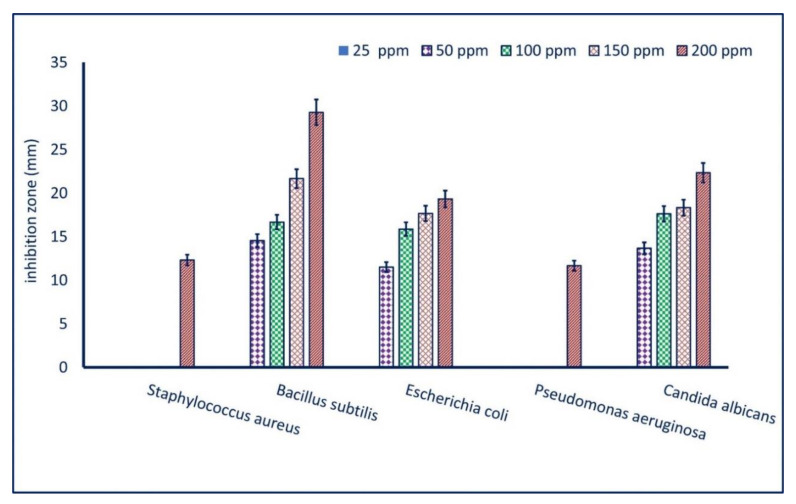
Antimicrobial activity of bacterially synthesized ZnO-NPs at different concentrations (200, 150, 100, 50, and 25 ppm) against Gram-positive bacteria (*B. subtilis* and *S. aureus*), Gram-negative bacteria (*E. coli* and *P. aeruginosa*), and unicellular fungi (*C. albicans*).

**Table 1 materials-14-06983-t001:** The efficacy of different concentrations (200, 150, 100, 75, 50, and 25 ppm) of bacterially synthesized ZnO-NPs as a larvicidal agent against third instar larvae of *Culex pipiens*.

Time	Mortality Percentages (%) at Different Concentrations (ppm)											
	Control (Zn(CH_3_COO)_2_·2H_2_O)						ZnO-NPs					
	25 ppm	50 ppm	75 ppm	100 ppm	150 ppm	200 ppm	25 ppm	50 ppm	75 ppm	100 ppm	150 ppm	200 ppm
24 h	0.0 ± 0.0	0.0 ± 0.0	10 ± 5.8	30 ± 0.0	43.3 ± 3.3	44.3 ± 3.3	30 ± 0.0	40 ± 5.8	50 ± 5.8	80 ± 5.8	85 ± 5.8	100 ± 0
48 h	0.0 ± 0.0	0.0 ± 0.0	23.3 ± 3.3	33.3 ± 3.3	46.7 ± 3.3	49.7 ± 3.3	36.7 ± 3.3	46.7 ± 3.3	56.7 ± 3.3	86.7 ± 3.3	90 ± 3.3	100 ± 0
72 h	0.0 ± 0.0	0.0 ± 0.0	30 ± 0.0	40 ± 0.0	53.3 ± 3.3	54.6 ± 3.3	40 ± 0.0	49 ± 0.0	60.7 ± 3.3	90 ± 0.0	94 ± 0.0	100 ± 0
96 h	0.0 ± 0.0	0.0 ± 0.0	33.3 ± 3.3	46.7 ± 3.3	63.3 ± 3.3	63.3 ± 3.3	46.7 ± 3.3	56 ± 0.0	67.3 ± 3.3	93.3 ± 3.3	96 ± 3.3	100 ± 0

## Data Availability

The data presented in this study are available on request from the corresponding author.

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
