# Peer review of "Green Synthesis of Zinc Oxide Nanoparticles (ZnO-NPs) by Pseudomonas aeruginosa and Their Activity against Pathogenic Microbes and Common House Mosquito, Culex pipiens"

_materials, 2021, doi:10.3390/ma14226983_

Round 1

Reviewer 1 Report

file attached

Author Response

Thank you very much for reviewing my manuscript, please see the attachment file for the author's response.

Reviewer 2 Report

The manuscript needs some work before it can be published. There are some missing data that should be presented in the manuscript.  Below are some questions/ comments for the authors to improve the manuscript.

What was the pH of the solution used to lyse the bacteria?

The authors do not under chemistry it is evident in the writing of the methodology, the authors wrote: “The metabolites secreted by bacterial strain NMJ15 were used as a reducing agent for zinc acetate (Zn(CH3COO)2.2H2O)..” There is no reduction going on in the synthesis.  The Zn in Zinc acetate is Zn(2+) and the Zn in ZnO is Zn(2+).  There is no reducing agent used in the manuscript.

The authors wrote: “…color change of CFF after being mixed with a precursor (Zn(CH3COO)2.2H2O) indicates the successful formation of ZnO-NPs..” this only indicates something happened doe not mean ZnO was formed, generally the formation of the white solution may indicate the formation of Zn hydroxide.

The data presented in the UV-Vis needs a better step rate, the data looks like the steps are too large which make the data look like segmented lines.

Something is wrong with the FTIR data there is a transmittance of approximately 130%. 

The diffraction pattern needs to be redone, longer counting times to enhance the signal.  In addition, the sample as precipitated should also be presented this would clarify the synthesis of the sample and at which point the ZnO is made.

Author Response

(The authors gave the same response as above.)

Reviewer 3 Report

The presented manuscript includes the green synthesis of zinc oxide nanoparticles (ZnO-NPs) by Pseudomonas aeruginosa and their activity against pathogenic microbes and common house mosquito, Culex pipiens.

Some general corrections and weak points should be mentioned:

Q1. One of the weakest points is the novelty. What a novelty of the presented study, e.g. comparing to https://doi.org/10.1016/j.jphotobiol.2016.12.011 Ahmed, S., Chaudhry, S. A., & Ikram, S. (2017). A review on biogenic synthesis of ZnO nanoparticles using plant extracts and microbes: a prospect towards green chemistry. Journal of Photochemistry and Photobiology B: Biology, 166, 272-284?

Q2. Line 21. Change “was” to “were” (nanoparticles).

Q3. Sentence “Data analyses showed that the activities were dose-dependent.” seems to be very obvious.

Q4. The second weakest point is: Why the mortality for Culex pipiens by the synthesized ZnO-NPs compared with Zn(NO3)2•6H2O? More reasonable would be to use ZnO-NPs synthesized by any other method.

Q5. Lines 54-57. There is another “green approach” known as solution combustion synthesis to obtain fine dispersed NPs including ZnO. E.g. this approach is mentioned in https://doi.org/10.1016/B978-0-12-821141-0.00013-6    Romanovski, V. (2021). Agricultural waste based-nanomaterials: Green technology for water purification. In Aquananotechnology (pp. 567-585). Elsevier.

Q6. Lines 332-333. “The other peaks observed in the XRD pattern…” – I can recommend to change the phrase. The reason is that high phone and wide peak/wave in the range 20-30 2theta “can be attributed to the crystallization of bacterial metabolites 333 including proteins and organic substances that coated the ZnO-NPs surface”.

Author Response

(The authors gave the same response as above.)

Round 2

Reviewer 2 Report

The reviewer still takes issue with some the experimentation and the writing of the results. The reviewer also fells that one experiment is missing from the work.  Below are comments for the authors.

The reviewer is unsure of why the authors do not understand there is no reduction of Zn occurring in the reaction mixture.  Zinc acetate contains or releases a Zn2+ ion into solution. The Zn2+ is then complexed to form Zn(OH)2 which is followed by a dehydration (loss of a water molecule) to form ZnO.  There is no metallic Zn present in the reaction. It is not possible or reasonable to assume the Zn2+ is recued to Zn(0) and re-oxidized to form Zn2+ coordinated to an oxygen. If there is a reduction then the authors should show the Zn metal intermediate by diffraction.

In order to say the bacteria or bacterial components are responsible for the formation of the ZnO nanoparticles, the authors need to show that it actually occurred.  IT would be recommended that the authors take the reaction mixture without the bacteria and add the zinc to see if the nanoparticles are formed.

The authors wrote the following “The reason for high phone and wide peak/wave in the range of 20-30 at 2 theta can be attributed to the crystallization of bacterial metabolites including proteins and organic substances that coated the ZnO-NPs surface as reported previously “  The reviewer is unsure of what the authors mane by the word phone. In addition, the diffraction pattern is of very poor quality.  The peaks are barely out of the rage of noise, and there is no peak width. The authors need to collect the diffraction pattern at a much longer counting time which will enhance the count rates of the signal and reduce the noise level. The current diffraction pattern the 002, 103, 201 are not distinguishable from the noise.  If we use the true definition of noise and signal (3 times the noise) then none of the peaks are actually above the noise. 

The authors write the following: “The XRD pattern (Figure 5) showed the intense diffraction peaks (100), (101), (102), (110), (103), (112), and (201) at 2θ values of 31.5°, 34.2°, 36.3°, 47.3°, 56.4°, 62.7°, 67.3°, and 96.1°.  Where did the 96.1 degrees come from? The last angle listed is past the diffraction angles recorded on the diffraction pattern.

Author Response

Thanks for helping to improve our manuscript, please see the attachment. 

Reviewer 3 Report

Accept in present form

Author Response

The authors would be like to thank the reviewer for his approval and acceptance.